# Impact of sex on use of low tidal volume ventilation in invasively ventilated ICU patients—A mediation analysis using two observational cohorts

**Pien Swart**[1]*, **Rodrigo Octavio Deliberato**[2,3], **Alistair E. W. Johnson**[4], **Tom J. Pollard**[4], **Lucas Bulgarelli**[4], **Paolo Pelosi**[5,6], **Marcelo Gama de Abreu**[7,8], **Marcus J. Schultz**[1,9,10,11], **Ary Serpa Neto**[1,2,12,13]

1 Department of Intensive Care, Amsterdam UMC, Amsterdam, The Netherlands, 2 Department of Critical Care Medicine, Hospital Israelita Albert Einstein, São Paulo, Brazil, 3 Big Data Analytics Group, Hospital Israelita Albert Einstein, São Paulo, Brazil, 4 Laboratory for Computational Physiology, Institute for Medical Engineering & Science, MIT, Cambridge, MA, United States of America, 5 IRCCS San Martino Policlinico Hospital, Genoa, Italy, 6 Department of Surgical Sciences and Integrated Diagnostics (DISC), University of Genoa, Genoa, Italy, 7 Pulmonary Engineering Group, Department of Anaesthesiology and Intensive Care Medicine, University Hospital Carl Gustav Carus, Technical University Dresden, Dresden, Germany, 8 Outcomes Research Consortium, Cleveland, OH, United States of America, 9 Laboratory of Experimental Intensive Care and Anaesthesia (L·E·I·C·A), Amsterdam UMC, Amsterdam, The Netherlands, 10 Mahidol Oxford Tropical Medicine Research Unit (MORU), Mahidol University, Bangkok, Thailand, 11 Nuffield Department of Medicine, University of Oxford, Oxford, United Kingdom, 12 Pulmonary Division, Cardio–Pulmonary Department, Hospital das Clinicas HCFMUSP, Faculdade de Medicina, Universidade de Sao Paulo, Sao Paulo, Brazil, 13 Australian and New Zealand Intensive Care Research Centre, Monash University, Melbourne, Australia

* p.swart@amsterdamumc.nl

**Data Availability Statement:** The two databases used in this study are owned by third parties. Access to the eICU Collaborative Research

## Abstract

### Background

Studies in patients receiving invasive ventilation show important differences in use of low tidal volume ($V_T$) ventilation (LTVV) between females and males. The aims of this study were to describe temporal changes in $V_T$ and to determine what factors drive the sex difference in use of LTVV.

### Methods and findings

This is a posthoc analysis of 2 large longitudinal projects in 59 ICUs in the United States, the 'Medical information Mart for Intensive Care III' (MIMIC III) and the 'eICU Collaborative Research DataBase'. The proportion of patients under LTVV (median $V_T$ < 8 ml/kg PBW), was the primary outcome. Mediation analysis, a method to dissect total effect into direct and indirect effects, was used to understand which factors drive the sex difference. We included 3614 (44%) females and 4593 (56%) males. Median $V_T$ declined over the years, but with a persistent difference between females (from median 10.2 (9.1 to 11.4) to 8.2 (7.5 to 9.1) ml/kg PBW) vs. males (from median 9.2 [IQR 8.2 to 10.1] to 7.3 [IQR 6.6 to 8.0] ml/kg PBW) (P < .001). In females versus males, use of LTVV increased from 5 to 50% versus from 12 to 78% (difference, −27% [−29% to −25%]; P < .001). The sex difference was mainly driven by patients' body height and actual body weight (adjusted average causal mediation effect, −30% [−33% to −27%]; P < .001, and 4 [3% to 4%]; P < .001).

Database starts with Completing the required CITI training course 'Data or Specimens Only Research' course. Follow the steps as mentioned on the website using the following link; https://eur04.safelinks.protection.outlook.com/?url=https%3A%2F%2Feicu-crd.mit.edu%2Fgettingstarted%2Faccess%2F&data=04%7C01%7Cp.swart%40amsterdamumc.nl%7C633426bc8a444650a08008d92cadf495%7C68dfab1a11bb4cc6beb528d756984fb6%7C0%7C0%7C637589948790335180%7CUnknown%7CTWFpbGZsb3d8eyJWIjoiMC4wLjAwMDAiLCJQIjoiV2luMzIiLCJBTil6Ik1haWwiLCJXVCI6Mn0%3D%7C2000&sdata=o1cyNvZo5bzsqhr5Bj7XOHlPyw6yLNUra9hWPUI0sgw%3D&reserved=0 On this website one can also see the steps needed to Request access to the eICU Collaborative Research Database after completing the course. It starts with registering for an account on PhysioNet. For the MIMIC III database it is more or less the same. Researchers seeking to use the database must formally request access with the steps mentioned on the website using the following link; https://eur04.safelinks.protection.outlook.com/?url=https%3A%2F%2Fmimic.physionet.org%2Fgettingstarted%2Faccess%2F&data=04%7C01%7Cp.swart%40amsterdamumc.nl%7C633426bc8a444650a08008d92cadf495%7C68dfab1a11bb4cc6beb528d756984fb6%7C0%7C0%7C637589948790345133%7CUnknown%7CTWFpbGZsb3d8eyJWIjoiMC4wLjAwMDAiLCJQIjoiV2luMzIiLCJBTil6Ik1haWwiLCJXVCI6Mn0%3D%7C2000&sdata=SOWjREhDJReRReDH%2BnlroNaw5fJihDIn5LZ0VvXLzBI%3D&reserved=0 The same training course as for the eICU database must be completed and also on this website you can see the steps that must be taken to request access. The authors had no special access privileges to the data others would not have.

**Funding:** The author(s) received no specific funding for this work.

**Competing interests:** Ary Serpa Neto reported receiving personal fees from Dräger outside of the submitted work. Marcelo Gama de Abreu reported receiving grants and personal fees from Drägerwerk AG and GlaxoSmithKline and receiving personal fees from GE Healthcare outside of the submitted work. No other disclosures were reported. This does not alter our adherence to PLOS ONE policies on sharing data and materials.

## Conclusions

While LTVV is increasingly used in females and males, females continue to receive LTVV less often than males. The sex difference is mainly driven by patients' body height and actual body weight, and not necessarily by sex. Use of LTVV in females could improve by paying more attention to a correct calculation of $V_T$, i.e., using the correct body height.

## Introduction

Before a landmark study regarding the benefit of so-called low $V_T$ ventilation (LTVV) was published in 2000 [1], it was common practice in critically ill patients to use a high tidal volume ($V_T$) during invasive ventilation, as this increased minute ventilation necessary to compensate for an often present disease–induced increase in dead space [2], and because this potentially reduced atelectases thereby reducing the need for high oxygen fractions ($FiO_2$) and positive end–expiratory pressure (PEEP) [3]. Convincing evidence for harm from ventilation with a high $V_T$ in patients with acute respiratory distress syndrome (ARDS) [1] resulted in worldwide introduction of LTVV often also in patients without ARDS. Indeed, a noticeable decline in $V_T$ size was seen in critically ill patients [4], albeit that many patients continued to receive ventilation with a too high $V_T$ [5, 6].

Often–cited challenges with LTVV include alleged increases in sedation need due to the higher respiratory rate necessary to compensate for a decline in minute volume with use of a low $V_T$. This could induce or worsen delirium, and may increase the risk for patient–ventilator asynchronies [7–9], although this was never confirmed in observational studies [10, 11]. Another practical problem with LTVV is that $V_T$ needs to be titrated against predicted body weight (PBW), which is a function of a patient's body height [1]––however, body height is often not known at the time of ICU admission, taking a patient's body height in a supine position can be difficult [12–16], and estimates of body height are frequently inaccurate in the ICU setting [13, 17–20]. Apart from this, doctors and nurses need to apply a not so easy equation at the bedside to calculate PBW, which could be challenging. For these reasons it has been proposed to use a 'default' $V_T$ [9]. This practice seemed to have been implemented widespread, as a 'rounded' absolute $V_T$ is often used [5, 6].

Two large observational studies, one in critically ill patients with ARDS [21], and one in patients receiving intraoperative ventilation during general anaesthesia for surgery [22], show that females receive LTVV much less often than males. We aimed to determine whether this sex difference in use of LTVV has narrowed over the years, we assessed the databases of 2 large longitudinal projects, the 'Medical Information Mart for Intensive Care III' (MIMIC–III) [23, 24] and the 'eICU Collaborative Research DataBase' (eICU) [25]. We also wished to determine which factors drive the sex difference. The aims were achieved, by testing the following hypotheses (1) that the difference in use of LTVV between females vs. males has narrowed, and (2) that the sex difference is mediated more by differences in patients' body height and weight, than by the use of a default $V_T$ or sex per se.

## Methods

For this analysis, data were retrieved from 2 conveniently–sized longitudinal projects, of which the databases contain granular data on ventilation management in ICU patients in the US between 2001 and 2015––the 'Medical Information Mart for Intensive Care III' (MIMIC–

III) v1.4 [23, 24], and the 'eICU Collaborative Research Database' (eICU) [25]. An elaborative description of these high temporal resolution databases can be found in the **S1 File**.

## Ethical approval

The Institutional Review Board of the Beth Israel Deaconess Medical Center (2001–P–001699/14) and the Massachusetts Institute of Technology (No. 0403000206) approved the MIMIC III project. Requirement for individual patient consent was waived because the project did not impact clinical care and all protected health information was deidentified. The eICU was exempt from institutional review board approval due to its retrospective design, lack of direct patient intervention, and the security scheme, for which the re–identification risk was certified as meeting the safety standards by Privacert (Cambridge, MA) (Health Insurance Portability and Accountability Act Certification no. 1031219–2).

## Patient selection

Patients were selected for inclusion in the current analysis if: 1) aged $\geq 16$ years; and 2) having had received invasive ventilation for at least 48 consecutive hours from start of ventilation. Patients were excluded if death occurred before 48 hours, and if ventilation was provided through a tracheostomy cannula at any time during the first 48 hours of ventilation. Patients who had incomplete datasets or datasets that did not sufficiently capture absolute $V_T$, body height or actual body weight (ABW) were also excluded. In patients who were admitted for ventilation more than one time, only data of the first ICU admission of the first hospitalization were used.

## Data extraction

Sex, body height and ABW data were extracted, in addition to other demographic data, including reason for and type of ICU admission, reason for invasive ventilation, and comorbidities. Body weight and height were collected from the medical records, i.e., the body weight and height as used for decisions in care. These could either be based on measurement at earlier contacts with a patient, measurements in the ward, or at admission to the ICU. Presence of ARDS was scored and classified, or re–classified when previous definitions were used, as 'mild', 'moderate' or 'severe' according to current Berlin definition for ARDS [26]. Ventilatory variables and parameters were extracted as the highest and the lowest values per time frame of 6 hours during the first 48 hours of ventilation. These values were summarized as the mean for every 6 hours and then as the median for the first 48 hours. $V_T$ was collected and normalized for ABW and PBW, as follows:

$$\text{VT, ABW} = \text{absolute VT [ml]}/\text{ABW [kg]} \tag{Eq 1}$$

And

$$\text{VT, PBW} = \text{absolute VT [ml]}/\text{PBW [kg]} \tag{Eq 2}$$

PBW was calculated as follows:

$$\text{PBW} = 45.5 + 0.90 * ([\text{body height in cm}] - 152.4)(\text{in females}) \tag{Eq 3A}$$

And

$$\text{PBW} = 50.0 + 0.90 * ([\text{body height in cm}] - 152.4)(\text{in males}). \tag{Eq 3B}$$

LTVV was defined as having had received a median $V_T$ over the first 48 hours $\leq$ 8 ml/kg PBW, a low $V_T$ was defined as $V_{T, PBW} \leq$ 8 ml/kg PBW. A patient was defined as possibly having had received ventilation with a default $V_T$ when the first two reported absolute $V_T$ were 'rounded numbers' i.e., a $V_T$ of 200, 250, 300, 350, 400, 450, 500, 550, 600 or 650 ml.

The primary outcome was use of LTVV. We also analysed which factors mediate the sex difference in use of LTVV.

## Statistical analysis

No assumptions were made for missing data. Descriptive statistics were reported for the study population stratified according to sex, and as number and relative proportions for categorical variables and median [IQR] for continuous variables. For baseline characteristics, the groups were compared using Fisher exact tests for categorical variables and Wilcoxon rank–sum test for continuous variables. For all analyses, males were used as the reference.

All analyses were performed using multilevel (patients nested in hospitals, within the datasets), mixed modelling with hospitals as random effect and dataset as a fixed effect. Heterogeneity between datasets was determined by fitting a fixed interaction term between sex and dataset, while overall sex effect was reported with dataset treated as a fixed effect and hospitals treated as a random effect.

Ventilatory variables and parameters where compared over time, and between males and females, and absolute differences with the respective 95%–confidence interval (CI) were calculated as the absolute difference from a mixed–effect linear model considering the dataset as a fixed effect and the year of admission and centres as random effect to account for within–centre and year clustering. Categorical variables were compared as the risk difference from the same model.

Cumulative distribution plots were used to plot the cumulative distribution frequency of ventilation variables and parameters, for males and females according to tertiles of year of admission. The mode, and median with interquartile range [IQR] of absolute $V_T$, $V_{T, ABW}$, and $V_{T, PBW}$ were calculated. $V_T$, $V_{T, ABW}$, and $V_{T, PBW}$ were further assessed according quintiles of body height and ABW.

Temporal shifts in body height, ABW, absolute $V_T$, $V_{T, ABW}$, and $V_{T, PBW}$ over the calendar years were compared between females and males using mixed–effect longitudinal models with patients and centres as random effect, and the dataset, year, sex and an interaction of year and sex as fixed effects. Two P–values were reported: 1) P–value for sex differences, reflecting the overall test for difference between sex across the years; and 2) P–values for the sex x year interaction, evaluating if change over time differed by sex. In addition, a P–value from a three–way fixed interaction between dataset, sex a year was reported to assess the consistency of the findings among the datasets assessed.

The proportion of patients having had received LTVV were described and an unadjusted mixed–effect generalized linear model was used to extract the risk difference among sex. To further assess if use of LTVV is associated with sex, even after adjustment for confounders, a mixed–effect multivariable model with centres and years as random effect was done. The model was adjusted by covariates according to clinical relevance and when unbalanced among the groups, namely: age, body height, ABW, Oxford Severity of Illness Score (OASIS), Sequential Organ Failure Assessment (SOFA), type of admission, pH, $PaCO_2$, and use of a titrated (i.e., non–default) $V_T$. In all models, continuous variables were standardized to improve convergence, and absolute differences represents changes according to one standard deviation increase in the predictor.

To investigate whether the difference in the use of LTVV between females and males are due to differences in body height, ABW or the use of a 'fixed' possibly sex–specific $V_T$, a mixed–effect multivariable mediation model was used. In a first step in the mediation analysis, we assessed the individual impact of body height, ABW and having had received a default $V_T$ as potential mediators for the difference in use of LTVV between sex in an univariable and multivariable model adjusted by all the covariates described above. For this model, Quasi–Bayesian 95% confidence intervals were estimated after 10,000 simulations. In a second step, body height, ABW, and having had received a default $V_T$ were included simultaneously in the model to assess the impact and importance of each. In this second model the confidence intervals were estimated from robust clustered standard errors. All mediation models included year and centres as random effect.

All analyses were conducted in R v.3.60 and a P–value $< 0.05$ was considered statistically significant.

**Posthoc analyses.**   We performed two posthoc analyses. The first posthoc analysis concerned patients who were admitted because of sepsis, and including patients with pneumo–sepsis. In this analysis we repeated the main analysis to compare temporal shifts in absolute $V_T$, $V_{T, ABW}$, and $V_{T, PBW}$ over the calendar years between females and males.

In a second posthoc analysis we explored the differences between sexes during controlled ventilation versus spontaneous ventilation. Also here we followed the main analysis.

# Results

## Patients

After exclusion of patients not meeting inclusion and exclusion criteria, a total of 8207 patients (44% females; 56% males) remained for the final analysis––3846 patients in the MIMIC III database, and 4361 patients in the eICU database (**Fig 1 in S1 File**). Females were slightly older, appeared sicker on admission, and more often had a history of chronic obstructive pulmonary disease (**Tables 1 and 2 in S1 File**). Females were shorter than males, and had a lower ABW. Body height did not change, but ABW increased over the years, in females and in males (**Fig 2 in S1 File**).

Median absolute $V_T$, median peak, plateau and driving pressure, median minute ventilation, median respiratory rate and median mechanical power declined over the years, in females and males (**Fig 1**, **Table 1**; **Table 3 in S1 File**). Females received ventilation with a default $V_T$ more often than males (**Fig 3 in S1 File**). While $V_{T, ABW}$ was similar in females versus males, $V_{T, PBW}$ remained persistently higher in females.

## Use of LTVV

LTVV was increasingly used over the years, in females and males (**Fig 2**). In females versus males, use of LTVV increased from 5 to 50% versus from 12 to 78% (difference, –27% [–29% to –25%]; P $< .001$).

## Mediation analysis

Results from the multivariate model are presented in **Table 2**; **Table 4 in S1 File**.

In the multivariable multiple mediation models, body height and ABW mediated the sex inequality in use of LTVV (**Table 3**). Use of default $V_T$ mediated the sex difference in use of LTVV in a model that only used this factor (P = 0.021), but not in the model using also patients' body height and ABW.

**Table 1. Ventilatory parameters in the first two days of ventilation**[*]**.**

| | Females | Males | Absolute Difference[**] | p value |
|---|---|---|---|---|
| | (n = 3614) | (n = 4593) | (95% CI) | |
| First day of ventilation | | | | |
| Tidal volume | | | | |
| Absolute, mL | 472 (412–518) | 550 (500–615) | -86.45 (-89.95 to -82.95) | < 0.001 |
| Mode | 500 | 600 | --- | --- |
| mL/kg PBW | 8.6 (7.7–9.7) | 7.6 (6.8–8.5) | 1.00 (0.93 to 1.07) | < 0.001 |
| Mode | 8.0 | 6.8 | --- | --- |
| mL/kg ABW | 6.4 (5.2–7.9) | 6.4 (5.3–7.6) | 0.05 (-0.02 to 0.13) | 0.161 |
| Mode | 8.0 | 6.1 | --- | --- |
| Use of default tidal volume | 1705 / 3468 (49.2) | 1965 / 4443 (44.2) | 4.93 (2.72 to 7.14) | < 0.001 |
| PEEP, cmH$_2$O | 5 (5–8) | 5 (5–8) | -0.19 (-0.31 to -0.07) | 0.002 |
| Respiratory rate, mpm | 18 (15–21) | 18 (15–21) | -0.01 (-0.20 to 0.17) | 0.886 |
| Plateau pressure, cmH$_2$O | 21 (17–25) | 20 (16–24) | 0.99 (0.73 to 1.25) | < 0.001 |
| Driving pressure, cmH$_2$O | 14 (12–18) | 13 (11–16) | 1.14 (0.92 to 1.36) | < 0.001 |
| Peak pressure, cmH$_2$O | 26 (21–30) | 25 (20–29) | 0.90 (0.61 to 1.19) | < 0.001 |
| Minute ventilation, L/min | 8.5 (7.0–10.1) | 10.0 (8.2–12.0) | -1.59 (-1.69 to -1.49) | < 0.001 |
| L/min PBW | 0.15 (0.13–0.19) | 0.14 (0.11–0.16) | 0.02 (0.02 to 0.02) | < 0.001 |
| Mechanical power, J/min | 15.0 (11.0–20.0) | 17.1 (12.4–23.5) | -2.71 (-3.04 to -2.39) | < 0.001 |
| J/min PBW | 0.27 (0.20–0.37) | 0.23 (0.17–0.33) | 0.04 (0.03 to 0.04) | < 0.001 |
| Ventilatory ratio | 1.90 (1.51 to 2.42) | 1.74 (1.41 to 2.19) | -0.18 (-0.24 to -0.14) | < 0.001 |
| Second day of ventilation | | | | |
| Tidal volume | | | | |
| Absolute, mL | 455 (400–507) | 550 (500–605) | -87.30 (-90.91 to -83.69) | < 0.001 |
| Mode | 500 | 600 | --- | --- |
| mL/kg PBW | 8.4 (7.6–9.6) | 7.5 (6.8–8.4) | 0.94 (0.87 to 1.01) | < 0.001 |
| Mode | 7.9 | 6.8 | --- | --- |
| mL/kg ABW | 6.3 (5.1–7.8) | 6.3 (5.2–7.5) | 0.03 (-0.05 to 0.10) | 0.492 |
| Mode | 10.0 | 6.1 | --- | --- |
| PEEP, cmH$_2$O | 5 (5–8) | 5 (5–9) | -0.28 (-0.41 to -0.15) | < 0.001 |
| Respiratory rate, mpm | 18 (15–21) | 18 (15–21) | -0.05 (-0.25 to 0.15) | 0.606 |
| Plateau pressure, cmH$_2$O | 21 (17–25) | 20 (17–24) | 0.70 (0.45 to 0.96) | < 0.001 |
| Driving pressure, cmH$_2$O | 14 (11–17) | 13 (10–16) | 0.95 (0.74 to 1.16) | < 0.001 |
| Peak pressure, cmH$_2$O | 25 (21–30) | 25 (20–29) | 0.31 (0.02 to 0.59) | 0.035 |
| Minute ventilation, L/min | 8.2 (6.8–10.0) | 9.9 (8.1–11.9) | -1.62 (-1.72 to -1.52) | < 0.001 |
| L/min PBW | 0.15 (0.12–0.18) | 0.13 (0.11–0.16) | 0.02 (0.01 to 0.02) | < 0.001 |
| Mechanical power, J/min | 14.5 (10.5–19.6) | 17.0 (12.3–23.8) | -3.11 (-3.45 to -2.77) | < 0.001 |
| J/min PBW | 0.26 (0.19–0.36) | 0.23 (0.16–0.33) | 0.03 (0.02 to 0.04) | < 0.001 |
| Ventilatory ratio | 1.75 (1.40 to 2.20) | 1.59 (1.28 to 1.98) | -0.15 (-0.19 to -0.12) | < 0.001 |

Data are median (quartile 25%–quartile 75%).

ABW: actual body weight; mpm: movements per minute; PBW: predicted body weight; PEEP: positive end–expiratory pressure.

[*] Data was aggregated as the median of 4 values per day (measured every 6 hours).

[**] Absolute difference calculated from a mixed–effect linear model with dataset and sex as fixed effect and hospitals as random effect.

## Posthoc analyses

The results of the posthoc analysis concerning 973 female and 1058 male patients with sepsis, did not change the findings, albeit that the differences did not reach statistical significance (**Fig**

**Table 2. Effect on low tidal volume ventilation\*.**

|  | Absolute Difference (95% CI) | *p* value |
| --- | --- | --- |
| Female sex | -0.06 (-2.84 to 2.70) | 0.964 |
| Age | -1.12 (-2.19 to -0.05) | 0.040 |
| Weight | -8.10 (-9.22 to -6.99) | < 0.001 |
| Height | 22.97 (21.50 to 24.44) | < 0.001 |
| OASIS | 1.40 (0.18 to 2.63) | 0.025 |
| SOFA | -0.63 (-1.85 to 0.60) | 0.314 |
| Type of admission |  |  |
| Elective surgery | 0 (Reference) | --- |
| Medical admission | 8.36 (4.85 to 11.88) | < 0.001 |
| Urgent surgery | 4.79 (-1.43 to 11.01) | 0.131 |
| pH | -0.11 (-1.28 to 1.05) | 0.847 |
| PaCO$_2$ | 5.80 (4.65 to 6.95) | < 0.001 |
| Use of titrated tidal volume | -4.30 (-6.39 to -2.19) | < 0.001 |

OASIS: Oxford acute severity of illness score; SOFA: Sequential Organ Failure Assessment.

\* Absolute difference calculated from a mixed–effect multivariable model with centers and years as random effect.

**4 in S1 File**). The second analysis in 1901 patients during controlled ventilation and 1190 patients during assisted ventilation illustrates that differences between the sexes were consistently present during controlled and spontaneous ventilation (**Fig 5 in S1 File**).

## Discussion

The results of this analysis of 2 longitudinal projects can be summarized as follows: (1) the proportion of females and males receiving LTVV increased over the last decades, but females remained at a higher risk of not receiving LTVV; (2) modus and median absolute $V_T$, $V_{T, ABW}$, and $V_{T, PBW}$ declined, both in females and males, but $V_{T, PBW}$ remained persistently higher in

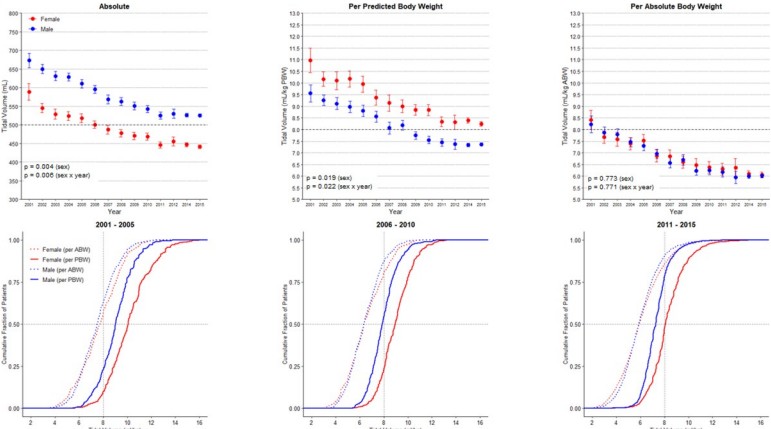

**Fig 1. Absolute tidal volume and corrected tidal volume (by predicted body weight or absolute body weight) over years and according to tertiles of year.** PBW: predicted body weight; ABW: actual body weight. *p* values for the sex reflect the overall test for difference between sex over the years while *p* values for the sex x year interaction evaluate if change over time differed by sex. There is no interaction between dataset and sex (*p* = 0.311, *p* = 0.100, *p* = 0.483 for absolute, corrected by PBW or ABW, respectively), or dataset and sex x year interaction (*p* = 0.310, *p* = 0.100, *p* = 0.482 for absolute, corrected by PBW or ABW, respectively).

**Table 3. Mediation analysis.**

|  | Adjusted Absolute Difference (95% CI)[a] | *p* value |
|---|---|---|
| **Univariable Mediation Model**[b] | | |
| Body height as mediator | | |
| Total effect of sex | -27.6 (-29.7 to -26.0) | < 0.001 |
| Average causal mediation effect of body height | -27.7 (-29.7 to -26.0) | < 0.001 |
| Average direct effect of female sex | 0.1 (-2.5 to 3.0) | 0.940 |
| Body weight as mediator | | |
| Total effect of sex | -27.6 (-29.7 to -26.0) | < 0.001 |
| Average causal mediation effect of body weight | 1.0 (0.4 to 2.0) | < 0.001 |
| Average direct effect of female sex | -28.6 (-30.7 to -26.0) | < 0.001 |
| Default $V_T$ as mediator | | |
| Total effect of sex | -27.7 (-29.8 to -26.0) | < 0.001 |
| Average causal mediation effect of titrated $V_T$ | 0.0 (-0.0 to 0.0) | 0.150 |
| Average direct effect of female sex | -27.7 (-29.9 to -26.0) | < 0.001 |
| Body height, weight and titrated $V_T$ as mediators* | | |
| Total effect of sex | -28.0 (-32.2 to -23.9) | < 0.001 |
| Average causal mediation effect of height | -30.7 (-34.2 to -27.2) | < 0.001 |
| Average causal mediation effect of weight | 3.3 (2.7 to 4.0) | < 0.001 |
| Average causal mediation effect of titrated $V_T$ | 0.0 (-0.1 to 0.1) | 0.818 |
| Average direct effect of female sex | -0.7 (-3.8 to 2.5) | 0.671 |
| **Multivariable Mediation Model**[c] | | |
| Body height as mediator** | | |
| Total effect of sex | -28.4 (-30.7 to -26.0) | < 0.001 |
| Average causal mediation effect of body height | -28.4 (-30.4 to -26.0) | < 0.001 |
| Average direct effect of female sex | -0.0 (-2.8 to 3.0) | 0.960 |
| Body weight as mediator*** | | |
| Total effect of sex | -0.3 (-3.1 to 3.0) | 0.820 |
| Average causal mediation effect of body weight | -0.3 (-0.8 to 0.0) | 0.310 |
| Average direct effect of female sex | -0.0 (-2.8 to 3.0) | 0.970 |
| Default $V_T$ as mediator**** | | |
| Total effect of sex | 0.1 (-2.6 to 3.0) | 0.939 |
| Average causal mediation effect of titrated $V_T$ | 0.2 (0.0 to 0.0) | 0.021 |
| Average direct effect of female sex | -0.0 (-2.8 to 3.0) | 0.968 |
| Body height, weight and titrated $V_T$ as mediators* | | |
| Total effect of sex | -27.0 (-31.3 to -22.7) | < 0.001 |
| Average causal mediation effect of body height | -30.1 (-33.5 to -26.6) | < 0.001 |
| Average causal mediation effect of body weight | 3.6 (2.9 to 4.4) | < 0.001 |
| Average causal mediation effect of titrated $V_T$ | 0.1 (-0.1 to 0.2) | 0.459 |
| Average direct effect of female sex | -0.7 (-3.9 to 2.5) | 0.679 |

[a] All estimated generated after 10,000 simulations if not otherwise indicated.

[b] Mixed–effect mediation model with Quasi–Bayesian confidence intervals, with hospitals and year as random effects and dataset as fixed effect.

[c] Mixed–effect mediation model with Quasi–Bayesian confidence intervals, with hospitals and year as random effects and dataset as fixed effect and adjusted for: age, OASIS, SOFA, type of admission, pH and $PaCO_2$.

* Confidence intervals estimated from robust clustered standard errors.

** Further adjusted by weight and titrated tidal volume.

*** Further adjusted by height and titrated tidal volume.

**** Further adjusted by weight and height.

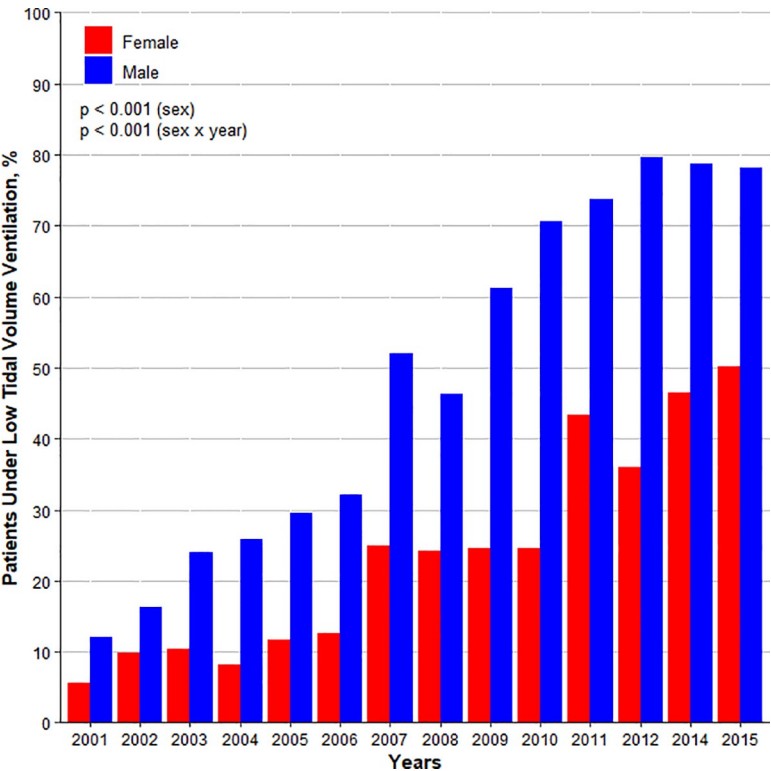

**Fig 2. Percentage of patients receiving low tidal volume ventilation.** Low tidal volume ventilation defined as a median of tidal volume in the first 48 hours of ventilation $\leq$ 8 ml/kg PBW. The overall difference on the use of low tidal volume ventilation among females and males corrected by the years is -26.7% (-28.8 to -24.6)%; $p < 0.001$. There is an interaction between dataset and sex ($p = 0.004$), and dataset and sex x year interaction ($p = 0.004$).

females; and (3) differences in use of LTVV between females and males were almost completely mediated by differences in body height and ABW between the sexes.

Strengths of this analysis are the use of 2 conventionally–sized high temporal resolution databases that contain granular ventilation data of two well–defined cohorts of patients from several centres from 2001 to 2015. In addition, all ventilation data comprised measurements realized every 6 hours in the first 48 hours of ventilation, instead of once a daily timepoints. This made it possible to analyse temporal changes and to better understand which factors are associated with $V_T$ titrations. The strict analysis plan, sophisticated statistical computations, and the mediation analysis made it possible to explain the difference in use of LTVV between females and males.

The finding that females receive ventilation with a higher $V_{T, PBW}$ than males is in line with results from previous investigations. Indeed, differences in median $V_T$ of ~1 to ~1.5 ml/kg PBW have been reported in general ICU populations [21, 27–29], as well as in specific groups like patients with ARDS [30] or with sepsis–related lung injury [31]. and in organ donors [32]. The same was found in patients receiving intraoperative ventilation [22, 33–37]. In patients undergoing prolonged elective abdominal surgery it were the females, patients with shorter stature or higher BMI that seemed more likely to receive large tidal volumes during general anaesthesia [33]. This association was also found in studies of patients undergoing intraoperative ventilation during general anaesthesia for surgery [34, 35]. The current study adds to these findings by showing that, the ICU sex inequality in use of LTVV is not a 'sex–issue', as differences in $V_T$ are driven by differences in patients' body height and weight and not by sex per se.

In the ICU setting, body height is frequently overestimated [17, 18], in particular in shorter patients [13, 19, 20]. In the here analysed cohorts, males were median 15 cm taller than females, which mirrors previous findings that worldwide males are on average >10 cm taller than females [5, 6, 31]. This difference could explain, at least in part, why females, who are thus usually shorter, are at a higher risk of not receiving LTVV [30, 31]. This is exactly what the mediation analysis suggests––body height is the most important mediator of the sex difference in LTVV.

A possible explanation could be a mismeasurement of body height. In ICU patients, ABW is frequently overestimated [13, 18–20], and it are the shorter patients in whom this happens most often [19]. The median difference in ABW between females and males of 14 kg in the here analysed cohorts, resembles findings in another study [18]. While $V_T$ should be titrated to PBW, and not to the ABW, this was often neglected [38, 39]. When $V_T$ is erroneously titrated to ABW, and if ABW is overestimated more often in shorter individuals, this could be another reason for why females receive LTVV less often than males. Yet, this is what was also suggested by the mediation analysis, albeit that it was a less important mediator than body height.

ICU patients frequently receive ventilation with a default $V_T$ [30]. The findings of the current analysis suggest that the proportion of females and males who possibly received ventilation with a default $V_T$ evolved in opposite ways, i.e., a decrease in females versus an increase in males. The reason for this remains unexplained. The benefits from using default $V_T$ are that this approach is more straightforward and easier to implement than performing a difficult calculation at the bedside. However, using a default $V_T$ has the potential to harm females more since their shorter body height makes that they are at greatest risk of receiving higher $V_T$ [40]. Nevertheless, use of default $V_T$ was associated with increased use of LTVV. Of note, the mediation analysis suggested that use of a default $V_T$ had a very small, actually negligible effect on the sex difference in use of LTVV.

There is abundant and convincing evidence that the use of an incorrect (i.e., too high) $V_T$ is associated with higher mortality and morbidity in ICU patients [21, 28, 29, 41–43]. In fact, $V_T$ is one of the key elements of lung–protective ventilation. Critically ill females tend to develop ARDS more often, and also have higher mortality rates [21, 28]. In addition, in female patients after cardiac surgery, a higher $V_T$ was found to have independent associations with organ dysfunction and prolonged ICU stay [29]. Whether the temporal changes in the current analysis translate in better outcomes, and whether this is different for females and males, cannot be reliably determined since we could not correct for temporal changes in other care processes that certainly have an impact on outcome as well. This analysis has a number of limitations. First, it was a posthoc analysis. Second, the present findings cannot be translated to patients ventilated through a tracheostomy canula or who died in the first 48 hours, since these patients were excluded from the current analysis. Only patients who received invasive ventilation for at least 48 hours were considered, this cohort will therefore consist of a selection of more severely ill patients, and may not generalize to the patients admitted to the ICU receiving invasive ventilation that are extubated within 48 hours. Third, the current analysis is the assumption that a rounded $V_T$ is a default $V_T$. Fourth, differences between the sexes in the posthoc analysis considering only patients with sepsis were not statistically significant. This, however, was probably to a lack of power since the number of patients used for this analysis was smaller. Of note, we only had information on ventilatory mode in the MIMIC-III database, as this was not collected in the eICU study.

## Conclusion

In these 2 large longitudinal projects spanning 2 decades of ventilation care for critically ill patients, we noticed a clear trend towards increased use of LTVV in females and males.

However, females continued to receive LTVV less often than males, a difference that seems mainly driven by differences in patients' body height and ABW, and not by sex or the use of a default $V_T$. These findings point out the importance to obtain a better insight into the practical challenges of titrating $V_T$ and the influence of these challenges on the use of LTVV.

## Supporting information

**S1 File. The supplement 'S1_File' contains all the supporting files for this submission.** (DOCX)

## Acknowledgments

We want to thank the team of the Laboratory for Computational Physiology from the Massachusetts Institute of Technology.

*Presentation*

*Sex Differences in Tidal Volume Limitation–an ICU database analysis*

Oral presentation at European Respiratory Society Annual International congress 2020, Online

## Author Contributions

**Conceptualization:** Pien Swart, Alistair E. W. Johnson, Marcus J. Schultz, Ary Serpa Neto.

**Data curation:** Ary Serpa Neto.

**Formal analysis:** Ary Serpa Neto.

**Methodology:** Pien Swart, Marcus J. Schultz, Ary Serpa Neto.

**Project administration:** Marcus J. Schultz, Ary Serpa Neto.

**Resources:** Marcus J. Schultz, Ary Serpa Neto.

**Software:** Ary Serpa Neto.

**Supervision:** Marcus J. Schultz, Ary Serpa Neto.

**Validation:** Ary Serpa Neto.

**Visualization:** Pien Swart.

**Writing – original draft:** Pien Swart, Marcus J. Schultz, Ary Serpa Neto.

**Writing – review & editing:** Pien Swart, Rodrigo Octavio Deliberato, Alistair E. W. Johnson, Tom J. Pollard, Lucas Bulgarelli, Paolo Pelosi, Marcelo Gama de Abreu, Marcus J. Schultz, Ary Serpa Neto.

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
