## [Decision Letter · Decision Letter 0]

13 Apr 2021

PONE-D-21-08610

Impact of Gender on Use of Low Tidal Volume Ventilation in Invasively Ventilated ICU Patients – a mediation analysis using two observational cohorts

PLOS ONE

Dear Dr. Swart,

Thank you for submitting your manuscript to PLOS ONE. After careful consideration, we feel that it has merit but does not fully meet PLOS ONE’s publication criteria as it currently stands. Therefore, we invite you to submit a revised version of the manuscript that addresses the points raised during the review process.

We look forward to receiving your revised manuscript.

Kind regards,

Aleksandar R. Zivkovic

Academic Editor

PLOS ONE

Journal Requirements:

Please provide additional details regarding participant consent. For each patient cohort, in the ethics statement in the Methods and online submission information, please ensure that you have specified (1) whether consent was informed and (2) what type you obtained (for instance, written or verbal, and if verbal, how it was documented and witnessed). If your study included minors, state whether you obtained consent from parents or guardians. If the need for consent was waived by the ethics committee, please include this information.

If your study includes an analysis of retrospectively collected data, please ensure that you have discussed whether all data/samples were fully anonymized before you accessed them and/or whether the IRB or ethics committee waived the requirement for informed consent. If patients provided informed written consent to have data/samples from their medical records used in research, please include this information.

Thank you for stating the following in the Acknowledgments/Funding Section of your manuscript:

Amsterdam University Medical Centers, location ‘AMC’, Amsterdam, The Netherlands

Thank you for stating the following in the Competing Interests section:

Ary Serpa Neto reported receiving personal fees from Dräger outside of the submitted work. Marcelo Gama de Abreu reported receiving grants and personal fees from Drägerwerk AG and GlaxoSmithKline and receiving personal fees from GE Healthcare outside of the submitted work. No other disclosures were reported.

We note that you received funding from a commercial source: Drägerwerk AG, GlaxoSmithKline, GE Healthcare

We noted in your submission details that a portion of your manuscript may have been presented or published elsewhere. „Gender Differences in Tidal Volume Limitation – an ICU database analysis. Oral presentation at European Respiratory Society Annual International congress 2020, Online”

Please clarify whether this conference proceeding or publication was peer-reviewed and formally published. If this work was previously peer-reviewed and published, in the cover letter please provide the reason that this work does not constitute dual publication and should be included in the current manuscript.

We note that you have indicated that data from this study are available upon request. PLOS only allows data to be available upon request if there are legal or ethical restrictions on sharing data publicly. For information on unacceptable data access restrictions, please see http://journals.plos.org/plosone/s/data-availability#loc-unacceptable-data-access-restrictions.

6a) If there are ethical or legal restrictions on sharing a de-identified data set, please explain them in detail (e.g., data contain potentially identifying or sensitive patient information) and who has imposed them (e.g., an ethics committee). Please also provide contact information for a data access committee, ethics committee, or other institutional body to which data requests may be sent.

6b) If there are no restrictions, please upload the minimal anonymized data set necessary to replicate your study findings as either Supporting Information files or to a stable, public repository and provide us with the relevant URLs, DOIs, or accession numbers. Please see http://www.bmj.com/content/340/bmj.c181.long for guidelines on how to de-identify and prepare clinical data for publication. For a list of acceptable repositories, please see http://journals.plos.org/plosone/s/data-availability#loc-recommended-repositories.

Please include captions for your Supporting Information files at the end of your manuscript, and update any in-text citations to match accordingly. Please see our Supporting Information guidelines for more information: http://journals.plos.org/plosone/s/supporting-information.

Reviewers' comments:

Reviewer #1: This is a pot joc analysis of two large longitudinal projects from 59 ICUs in USA including 8207 patients with mechanical ventilation >48 hours . Temporal changes of Vt and gender differences were analyzed during the period 2001-2015 . The authors did not included more recent data (2015-2020) with probably higher implementació of protective ventilation . Vt declines over the years in both sex but with a persistent difference between females (Vt 10.2 to 8.2) vs males ( Vt 9.2 to 7.3 ) ml/Kg PBW. What is the real clinical impact and relevance of a gender difference of median Vt 0.9 ml/KgIBW among both groups ? What is the Vt atributable mortality considering that the majority of respiratory mechanical parameters were in the recommended level of protective mechanical ventilation and no gender differences were observed except for Minute ventilation ? What was ventilatory ratio in both groups ? Additional Comments :

1. How and when actual body weigth and heigth were messured ?

2. The authors used a hight resolution data base . Could you please explain what you mean by high resolution ? What is the quality control of the data base ?

3. Table S2. Body weigth should be vidu heigth

4. Sepsis including pneumònia was the most frequent Initial diagnosis that increases overtime associated with a lower Pa/FiO2 . What was the gender Vt difference in this group of patients with probably a lower lung compliance ?

Reviewer #2: The authors report an interesting and clinically relevant study in a retrospectively mode on the use of low tidal volume ventilation (LTVV) in female versus male patients over the past years. The main finding was, that the use of LVVT improved over years, but less consequent in women compared to men, but this finiding is more due to the miscalculation of body height than of the gender per se.

The study is adequately performed in design, statistics and presentation of results. The findings are important and should be recognized by intensivists. I have just a few minor comments:

1. abstract p4, l 88: it should read "...using the correctly assessed body height for calculation of ideal body weight."

2. Introduction, p 5, l 91: "in the past..." is not correct, since the ARDSNet study is now 21 years old, may be it could read: "Although a landmark study regarding the benefit of LTVV was published in 2000, int hte past..."

3. p 6, l117-118: misphrase"...over recent years over the years..."

4. the references are mostly unstructured and not formatted to journals style

5. p7, l 140-150: Since the variation of VT is very dependent from the mode of MV (controlled vs. augmented spontaneously) the authors should add in a senternce which ventiklation mode was used in the early phase of MV in the studies.

6. PLOS authors have the option to publish the peer review history of their article (what does this mean?). If published, this will include your full peer review and any attached files.

Reviewer #1: No

Reviewer #2: **Yes: **Prof. Dr. Thomas Bein, MD

---

## [Author Response · Author response to Decision Letter 0]

14 Jun 2021

A point–to–point reply to your remarks and the reviewers are included in the cover letter. File name ' Response to Reviewers, including cover letter'

---

## [Editor Report · Decision Letter 1]

16 Jun 2021

Impact of sex on use of low tidal volume ventilation in invasively ventilated ICU patients - A mediation analysis using two observational cohorts

PONE-D-21-08610R1

Dear Dr. Swart,

We’re pleased to inform you that your manuscript has been judged scientifically suitable for publication and will be formally accepted for publication once it meets all outstanding technical requirements.

Kind regards,

Aleksandar R. Zivkovic

Academic Editor

PLOS ONE

---

## [Editor Report · Acceptance letter]

1 Jul 2021

PONE-D-21-08610R1 

Impact of sex on use of low tidal volume ventilation in invasively ventilated ICU patients – A mediation analysis using two observational cohorts 

Dear Dr. Swart:

I'm pleased to inform you that your manuscript has been deemed suitable for publication in PLOS ONE. Congratulations! Your manuscript is now with our production department. 

Kind regards, 

on behalf of

Dr. Aleksandar R. Zivkovic 

Academic Editor

PLOS ONE